# Characterization of a Dye-Decolorizing Peroxidase from *Irpex lacteus* Expressed in *Escherichia coli*: An Enzyme with Wide Substrate Specificity Able to Transform Lignosulfonates

**DOI:** 10.3390/jof7050325

**Published:** 2021-04-22

**Authors:** Laura Isabel de Eugenio, Rosa Peces-Pérez, Dolores Linde, Alicia Prieto, Jorge Barriuso, Francisco Javier Ruiz-Dueñas, María Jesús Martínez

**Affiliations:** Centro de Investigaciones Biológicas Margarita Salas (CIB), Consejo Superior de Investigaciones Científicas (CSIC), Ramiro de Maeztu 9, 28040 Madrid, Spain; lidem@cib.csic.es (L.I.d.E.); romapeces@gmail.com (R.P.-P.); lolalinde@cib.csic.es (D.L.); aliprieto@cib.csic.es (A.P.); jbarriuso@cib.csic.es (J.B.); fjruiz@cib.csic.es (F.J.R.-D.)

**Keywords:** fungi, oxidoreductases, DyP, lignocellulosic biomass, lignin

## Abstract

A dye-decolorizing peroxidase (DyP) from *Irpex lacteus* was cloned and heterologously expressed as inclusion bodies in *Escherichia coli*. The protein was purified in one chromatographic step after its in vitro activation. It was active on ABTS, 2,6-dimethoxyphenol (DMP), and anthraquinoid and azo dyes as reported for other fungal DyPs, but it was also able to oxidize Mn^2+^ (as manganese peroxidases and versatile peroxidases) and veratryl alcohol (VA) (as lignin peroxidases and versatile peroxidases). This corroborated that *I. lacteus* DyPs are the only enzymes able to oxidize high redox potential dyes, VA and Mn^+2^. Phylogenetic analysis grouped this enzyme with other type D-DyPs from basidiomycetes. In addition to its interest for dye decolorization, the results of the transformation of softwood and hardwood lignosulfonates suggest a putative biological role of this enzyme in the degradation of phenolic lignin.

## 1. Introduction

Lignin is the most abundant polymer in land ecosystems, after cellulose. Both of them, together with hemicelluloses, constitute the lignocellulosic biomass. Lignin´s recalcitrant nature protects cellulose from microbial breakdown [1]. This heterogeneous polymer is formed by random polymerization of *p*-coumaryl, coniferyl, and sinapyl alcohols [2], forming a complex three-dimensional network that can only be modified by oxidoreductases secreted by microorganisms, mainly white rot basidiomycetes [3]. The arsenal of oxidoreductases in these organisms is the main driver of lignin degradation, leaving a whitish material enriched in cellulose and hemicellulose [4]. Peroxidases, laccases, oxidases, dehydrogenases, monooxygenases, and peroxygenases are some of the oxidoreductases secreted by fungi for biodegradation of lignin and other recalcitrant compounds [5]. Peroxidases differ in overall fold and enzymatic activities but share a heme molecule as prosthetic group that is ligated by a histidine or cysteine.

Dye-decolorizing peroxidases (DyPs, EC 1.11.1.19) are a new family of heme peroxidases with substrate preference for xenobiotic azo- and anthraquinone-type dyes that are not good substrates for other peroxidases [6]. DyPs have been classified according to sequence similarity into four types, A, B, C, and D [7]. Type D DyPs, exclusively found in fungi, share tertiary structures (a ferredoxin-like fold) with classes A, B, and C, although they do not share high sequence similarity. In the active site, a proximal histidine acts as a strong iron ligand, and an arginine is found in the heme cavity, while distal histidine (present in plant peroxidases) is absent in DyPs—its place is occupied by a conserved aspartate (in some cases glutamate), included in the DyP-typifying sequence fingerprint motif GXXDG.

An attractive characteristic of DyPs is their resistance to elevated temperatures, pressures, and acidic conditions [8]. The catalytic cycle of DyP-type enzymes starts with their activation by peroxide. Then, it carries out two consecutive monovalent oxidations on different substrates before returning to the resting state (Figure 1).

There are a high number of putative DyP sequences deposited in protein databases that belong to many bacterial and fungal phyla, though only a few have been characterized [10]. Among them, those from the fungi *Bjerkandera adusta* and *Auricularia auricula-judae* have been biochemically characterized and crystallized [11,12,13,14]. DyP genes are frequent in the so-called wood-decay fungi, being more common in the genomes of white-rot (ligninolytic) than in brown-rot species [10]. This fact, together with their ability to oxidize the dimers of both non-phenolic [13,15] (in much lower extent than LiPs and VPs) and phenolic lignin models, suggests that fungal DyPs may contribute to lignin biodegradation [16].

Classic ligninolytic peroxidases can be classified into: (i) lignin peroxidases (LiP; EC 1.11.1.14), able to directly oxidize non-phenolic lignin model compounds, such as veratryl alcohol, through a solvent-exposed tryptophyl radical that transfers the electrons to the heme by a long-range electron transfer (LRET) pathway [3,17]; (ii) manganese peroxidases (MnP; EC 1.11.1.13), which oxidize Mn^2+^ in a site formed by three acid residues and the internal heme propionate, with the resulting Mn^3+^ acting as a diffusible redox mediator able to oxidize lignin phenolic units [18]; and (iii) versatile peroxidases (VP; EC 1.11.1.16) that combine the catalytic properties of LiP and MnP because they present both a Mn^2+^ oxidation site and a surface catalytic tryptophan for the oxidation of non-phenolic compounds [3]. These enzymes are characterized by their high redox potential, oxidizing the lignin polymer and other phenolic and non-phenolic aromatic compounds [18].

*Irpex lacteus* is a white-rot basidiomycete with potential biotechnological interest that has been applied in biodegradation of toxic compounds, dye decolorization, water and soil bioremediation, and biopretreatment of lignocellulosic substrates [19,20]. In previous studies, the cpop21 peroxidase from *I. lacteus* was identified as a putative DyP-type enzyme [19]. After determining its catalytic properties, its effect in wheat straw saccharification was tested, showing a significant increase in the recovery of fermentable glucose that indicates its synergistic action with cellulases [21].

The biological role of DyPs in nature is, however, not clear. This is due in part to the complicated structure of its substrate since initial lignin oxidation rates cannot be measured under steady-state conditions. For that reason, several lignin preparations, as lignosulfonates, are used in many instances to mimic the behavior of oxidoreductases in lignin oxidation [22]. These molecules are water-soluble low molecular weight compounds originated by sulfite pulping of wood. As a side effect, their polymerization can improve their biotechnological use as dispersants [23].

In this work we report the cloning and expression in *Escherichia coli* of *I. lacteus* DyP, which showed promising properties for its application in biotechnological processes [24]. The production and properties of the recombinant protein (IrlacDyP) are discussed and compared with those from the native enzyme. Furthermore, the ability of the recombinant protein to oxidize lignosulfonates was evaluated as an example of a biotechnological application of interest.

## 2. Materials and Methods

### 2.1. Gene Sequencing and Sequence Analysis

N-terminal sequencing and peptide mass fingerprinting analyses of IrlacDyP were previously performed [21]. Protein sequences of the DyPs displaying higher identity to IrlacDyP (and their coding nucleotide sequences (mRNA)) were retrieved from the database, and the nucleotides coding for the predicted signal peptide were removed after analysis using the SignalP 4.1 server [25]. The ClustalW alignment of the sequences allows the identification of conserved regions that were used to design degenerated primers for PCR amplification, along with the N-terminus and the internal peptide QLVPEFHK. The primers obtained were as follow: IrlacDyP N-term 5′-WSYGCSGGSAAYGAYAGCCTTC-3′, IrlacDyP QLVPEFHK Fw 5′-CAGCTKGTNCCNGAATTYCAYAAR-3′, IrlacDyP QLVPEFHK Rv 5′-YTTRTGRAATTCNGGNACMAGCTG-3′, IrlacDyP 1314 5′-GATAGADGGMRMGAARAARTATT-3′, and IrlacDyP 1309 5′-TCACGCTCCAAAATGCTCTACGAGAGCTGTAATAGA-3′ (stop codon underlined).

*I. lacteus* was cultivated in corn steep solids-based (CSS) medium [26] at 28 °C and 180 rpm for 5 days. The growth medium contained, per L: corn steep solids (26.3 g), glucose (40 g), FeSO_4_·7H_2_O (0.4 g), (NH_4_)_2_SO_4_ (9 g), KH_2_PO_4_ (4 g), and CaCO_3_ (7 g). RNA was extracted using the “RNeasy Plant Mini Kit” (Qiagen, MD, USA), including the DNAse treatment specified by the manufacturer. Reverse transcription of RNA was done using the “GeneAmp RNA PCR Reagent Kit” (Applied Biosystems, MA, USA). The PCR amplifications with the different primers’ combinations were carried out in a Mastercycler Pro S (Eppendorf, Hamburg, Germany) in a final volume of 50 µL with: 1× PCR Taq buffer, 2.5 U of Taq polymerase (Invitrogen, MA, USA), 15 mM MgCl_2_, 0.25 mM of each primer, and 1 mM of dNTPs (New England Biolabs, MA, USA). One hundred nanograms of cDNA were used as templates. Cycling parameters were 94 °C for 3 min, followed by 30 cycles of 94 °C for 45 s, 55–60 °C for 45 s, and 72 °C for 1.5 min and a final extension of 10 min at 72 °C. PCR products were run in a 1% agarose gel and subsequently purified and inserted into pGEM-T easy cloning vector (Promega, WI, USA). After transformation of the recombinant vectors into the *E. coli* DH5α strain, 3 clones containing the inserted fragments were sequenced using the BigDye Terminator v3.1 Cycle Sequencing kit and the automated ABI Prism 3730 DNA sequencer (Applied Biosystems, MA, USA). Sequences were assembled using Bioedit 7.1.3 software (North Carolina State University, NC, USA), and the IrlacDyP coding sequence from mRNA without signal peptide was deposited in GeneBank under accession number KR632636.

The glycosylation of the mature protein was analyzed using the NetNGlyc 1.0 and NetOGlyc 4.0 servers. Its evolutionary history was inferred by using the maximum likelihood method and Whelan And Goldman + Freq. model [27]. The tree with the highest log likelihood (−45,189.34) is shown. Initial tree(s) for the heuristic search were obtained automatically by applying neighbor-joining and BioNJ algorithms to a matrix of pairwise distances estimated using the JTT model and then selecting the topology with superior log likelihood value. The tree is drawn to scale, with branch lengths measured in the number of substitutions per site. This analysis involved 68 amino acid sequences. There was a total of 971 positions in the final dataset. Evolutionary analyses were conducted in MEGA X [28].

A three-dimensional model of the protein was generated using the programs implemented by the automated protein homology-modeling server SWISS-MODEL (Swiss Institute of Bioinformatics, Laussane, Switzerland) [29]. The structures with higher homology to the *I. lacteus* sequence were DyPs from *B. adusta* (59% identity) (PDB 3MM3) and *A. auricula-judae* (52% identity) (PDB 4UZI). The models were analyzed using PyMol 1.1 (http://pymol.org/).

### 2.2. Expression of Irpex lacteus DyP

To optimize the expression of the enzyme, the coding sequence codons were optimized and synthesized by ATG:biosynthetics (Merzhausen, Germany) (GeneBank KR632637). The restriction targets EcoRI at 5′ and NotI at 3′ were used for its introduction into the vector pET28a(+) (Novagen). The obtained pET28a(+):IrlacDyP vector was transformed into chemocompetent *E. coli* Bl21(DE3) pLysS cells [30].

For recombinant production of the *I. lacteus* DyP, *E. coli* cells containing the pET28a(+): IrlacDyP vector were grown overnight at 37 °C and 180 rpm in 500 mL flasks containing 200 mL of Luria-Bertani broth (LB) supplemented with 50 µg/mL of kanamycin and 34 µg/mL of chloramphenicol. The precultured cells were used to inoculate 2 L flasks containing 0.5 L of LB supplemented with kanamycin and chloramphenicol, which were grown for 2 h at 37 °C and 200 rpm (OD_600nm_ 0.6). The cultures were induced with 1 mM IPTG (isopropyl β-D-thiogalactoside), grown for 4 h and then harvested by centrifugation. Protein expression was monitored by sodium dodecyl sulfate-polyacrylamide gel electrophoresis (SDS-PAGE). For that, the cells from 1 mL *E. coli* cultures were harvested, resuspended in 50 µL of loading buffer, and boiled. SDS-PAGE was performed in 10% polyacrylamide gels using dual color precision plus protein (Bio-Rad) as standard and Coomassie R-250 staining. The bacterial pellet, corresponding to 5 L of culture, was resuspended in 50 mL lysis buffer containing 50 mM Tris–HCl (pH 8.0), 10 mM ethylenediaminetetraacetic acid (EDTA), and 5 mM dithiothreitol (DTT), and the mixture was incubated with 2 mg/mL lysozyme for 1 h on ice. Then 0.1 mg/mL of DNaseI was added and incubated for another 30 min. The solution was sonicated (1 min for three times with intervals on ice) and centrifuged 1 h at 15,000× *g*. The pellet containing the DyP polypeptide inclusion bodies was washed with 20 mM Tris-HCl (pH 8.0) containing 1 mM EDTA and 5 mM DTT. The protein was solubilized in 10 mL of 50 mM Tris-HCl (pH 8.0) containing 8 M urea, 1 mM EDTA, and 5 mM DTT, gently stirred for 1 h at 4 °C to complete DyP solubilization and centrifuged for 15 min at 15,000 × *g* to eliminate insoluble debris.

Optimal conditions for folding and activation of IrlacDyP were determined by performing a multifactorial approximation in 96-well plates using 200 μL of the folding mixture. The variables analyzed were: pH (using Britton–Robinson buffer at 6.0–9.5), oxidized L-glutathione (GSSG) (0–1.6 mM), urea (0.16–1.2 M), hemin (10 and 20 μM), and temperature (4 and 25 °C). The concentrations of protein (0.115 mg/mL), DTT (0.02 mM), and EDTA (0.1 mM) were kept constant. The enzyme activity against 2,2′-Azino-bis(3-ethylbenzothiazoline-6-sulfonic acid) diammonium salt (ABTS) was measured under standard conditions for multiwell plates every 24 h for 7 days. Prior to quantifying the activity, poorly folded proteins were removed by centrifugation at 2200× *g*, 4 °C, 10 min.

The in vitro folding was scaled-up in a final volume of 3 L using the optimal parameters defined in the 96-well plate. The activity of the folding mixture was followed spectrophotometrically against 1.25 mM ABTS and 50 μM Reactive Blue 19 (RB19) in 80 mM sodium tartrate buffer pH 3.0 in the presence of 0.4 mM H_2_O_2_ in a 0.5 ml UV-visible cuvette. After the incubation period, the folding mixture was concentrated by tangential filtration (Pellicon and Amicon, Millipore) using a 10 kDa cut off membrane (Membrane Cassete, Millipore). The material was then dialyzed in 10 mM sodium acetate buffer pH 4.3 for 16 h to precipitate free hemin. Insoluble hemin and those proteins that were not well folded were removed by centrifugation at 4200 × *g* at 4 °C for 30 min. The supernatant obtained was dialyzed for 16 h in 20 mM sodium acetate buffer pH 5.5 for subsequent purification.

IrlacDyP was purified by anion-exchange chromatography in a 6 mL Resource Q column (GE Healthcare, IL, USA) coupled to an ÄKTA purifier fast protein liquid chromatography system. The activated enzyme obtained as described above was loaded onto the column in 20 mM sodium acetate buffer pH 5.5 (2 mL/min). Non-bound proteins were washed using 14 mL of the same equilibration buffer. Then, retained proteins were eluted with a NaCl gradient from 0% to 50% elution buffer (20 mM sodium acetate, pH 5.5, 1 M NaCl) in 40 mL. The column was finally washed with 10 mL elution buffer and equilibrated to initial conditions. Proteins’ elution was followed at 280 and 410 nm as indicators of total and heme-containing proteins, respectively. Selected fractions, containing purified IrlacDyP, were dialyzed with equilibration buffer to remove NaCl. Purification was monitored by measuring ABTS oxidation, under standard conditions, of the collected fractions.

Native IrlacDyP was produced from *I. lacteus* cultures and purified as described in Salvachúa et al. [21].

Protein concentration was determined either using a Nanodrop 2000 (Thermo Fisher Scientific, MA, USA), the Pierce™ BCA ProteinAssay Kit (Thermo Fisher Scientific, MA, USA), or the QubitTM 3.0 Fluorometer (Thermo Fisher Scientific, MA, USA).

### 2.3. Recombinant Irpex lacteus DyP Characterization

IrlacDyP homogeneity was analyzed by SDS-PAGE, as described above.

To calculate its isoelectric point, the protein was loaded onto a 5% native polyacrylamide gel, including ampholytes, in a pH range 3.0–10.0 (GE Healthcare, IL, USA) and subjected to vertical electrophoresis using 7 mM H_3_PO_4_ as anode buffer and 50 mM NaOH as cathode buffer. After running, the pH gradient in the gel was determined by measuring the pH of the gel with a surface electrode. Then, the gel was washed twice in 100 mM sodium tartrate pH 3 for 5 min, and the peroxidase activity of active DyP was detected in a zymogram, using 1.25 mM ABTS and 10 mM H_2_O_2_ in 100 mM sodium tartrate buffer pH 3.0. Visible green bands were detected after 1 min incubation. The pI of the protein was calculated, taking into account the pH calibration line of the gel.

The UV-visible spectrum of IrlacDyP in 20 mM sodium acetate buffer pH 5.5 (270–700 nm range) was analyzed to ascertain its oxidation state and the coordination state and environment of the heme group. The Reinheitszahl (Rz) value was calculated as the ratio of the heme Soret band absorbance (at 403 nm) to that of the protein (at 280 nm). In addition, the heme content was determined by the pyridine-hemochrome method, which allows the empirical estimation of the molar extinction coefficient of the enzyme [31].

The formation of the transient states of the peroxidase catalytic cycle can be studied by the changes undergone by their spectroscopic properties when passing from the resting state (RS) to the compound I (CI) and compound II (CII) activated states. To analyze the peroxidase catalytic cycle, the enzyme was activated (~2.48 nM) with 5 H_2_O_2_ equivalents (~12.4 nM) in 20 mM sodium acetate at two pHs (3.0 and 5.5), and the spectral changes were measured in a wavelength range of 300–750 nm in a diode-array detector spectrophotometer for 30 min, measuring the total spectrum at 5 s intervals.

Circular dichroism spectroscopy measurements were carried out in a JASCO J-720 spectropolarimeter. Far-UV spectra were recorded in a 0.1 cm path length quartz cell at a protein concentration of 0.1 mg/mL in 10 mM sodium tartrate buffer, pH 5.0. Five consecutive scans were accumulated, and the average spectra were stored. The data were corrected for the baseline contribution of the buffer, and the observed ellipticities were converted into the mean residue ellipticities (θ), based on a mean molecular mass per residue of 110 Da.

Optimum pH of the protein was studied against various substrates. When ABTS (1.25 mM), RB19 (50 μM), Reactive Black 5 (RB5, 25 μM), veratryl alcohol (VA, 10 mM), and 2,6-dimethoxyphenol (DMP, 2.5 mM) where assayed, Britton-Robinson buffer 50 mM was used in a pH range of 1.5–9.0. For MnSO_4_ (6 mM), 100 mM sodium tartrate buffer was used at pH 3.1–5.8. H_2_O_2_ (0.4 mM) was added in all reactions. The results are expressed as the percentage of oxidation or discoloration, as an average of three replicates, and considering the maximum of activity against each substrate as 100%.

The pH stability at two temperatures (4 °C and 25 °C) was tested by incubating the enzyme (~50 nM) in 100 mM Britton-Robinson buffer in a pH range of 2.0–9.0 in the presence of 0.04% BSA (Thermo Fisher Scientific). Peroxidase activity was measured in triplicate in a 96-well plate after 0, 1, 2, 4, 6, 24, 48, 72, 96, and 168 h under standard conditions for ABTS, and the result is expressed as a percentage.

The thermostability index (T_50_), defined as the temperature at which 50% of activity is lost after 10 min incubation, was determined incubating the protein (~25 nM) in the presence of 0.04% BSA at its most stable short-term pH (10 mM sodium tartrate buffer, pH 3.0) at different temperatures between 30–65 °C for 10 min. The samples were incubated on ice for 5 min, tempered for 5 min, and the residual activity was measured by three replicates under standard conditions against ABTS.

Inactivation of the protein by H_2_O_2_ was measured using different molar equivalents (H_2_O_2_)/(enzyme). The enzyme concentration was kept constant (75 nM), and different concentrations of hydrogen peroxide (20–25,000 μM) in 100 mM sodium tartrate buffer pH 3.0 at 4 °C were used. Samples were taken at various times (5, 10, 20, 40, 60 min), and residual activity was measured in triplicate in a 0.5 mL UV visible cuvette under standard conditions for ABTS.

The *K*_m_ (Michaelis-Menten constant) is defined as the amount of substrate needed to reach half of the maximum reaction rate, and the *k*_cat_ (catalytic constant) is defined as the maximum reaction rate at a constant enzyme concentration. The ratio of *k*_cat_ to *K*_m_ (catalytic efficiency) indicates the number of molecules per substrate that are converted into product per second. The kinetic constants (*K*_m_, *k*_cat_, and *k*_cat_/*K*_m_) were calculated based on the absorbance changes produced by the oxidation of different substrate concentrations. The values of each constant and their corresponding errors were calculated using the Sigma Plot 12.5 program. Mean values and standard errors for affinity constant (*K*_m_) and enzyme turnover (*k*_cat_) were obtained by nonlinear least-squares fitting of the experimental measurements to the Michaelis-Menten model. The catalytic efficiency was calculated by fitting the experimental data to the normalized equation v = (*k*_cat_/*K*_m_)[S]/(1 + [S]/*K*_m_).

The concentration of H_2_O_2_ was kept constant (0.4 mM) when measuring the oxidation of ABTS, RB19, RB5, DMP, VA, and MnSO_4_, while the concentration of ABTS was kept constant (1.25 mM) to measure the kinetic constants for enzyme activation by H_2_O_2_. The reactions were carried out at the optimum pH determined for each substrate, and the temperature was kept constant at 25 °C. The different conditions used to perform the enzymatic kinetics of each substrate are summarized in Table 1.

### 2.4. Lignosulfonates Oxidation by IrlacDyP

Softwood (*Picea abies*) and hardwood (*Eucalyptus grandis*) lignosulfonates (LS), supplied by Borregaard-LignoTech, were used as substrate. First, they were dialyzed in 50 mM Tris-HCl (pH 8.0) buffer containing 10 mM EDTA to remove traces of Mn^2+^. Then they were dialyzed in bi-distilled water to remove the buffer and finally lyophilized [32] and resuspended in double-distilled water at a concentration of 20 g/L. The reaction mixture contained LS at 5 g/L, IrlacDyP (0.25 μM and 0.50 μM), and 100 mM sodium tartrate buffer pH 3.0. The reaction was carried out at 25 °C. A total of 14,000 total molar equivalents of H_2_O_2_ (3.5 mM or 7 mM, respectively) was added in 7 pulses of 0.5 mM or 1 mM during 24 h. Phenolic content was determined using Folin-Ciocalteu reagent (Merck, NJ, USA), adapting the protocol described to measure the reduction of these compounds in a 96-well plate [23]. This method is based on colorimetric detection of phenolic content. Free phenol’s radicals produced by an oxidative activity on the lignosulfonates would couple and polymerize into lignosulfonates with more homogenous size, larger molecular weight, and lower phenolic content [33,34]. In a 96-well plate, 5 μL of treated LS was mixed with 15 μL of Folin-Ciocalteu reagent, and bi-distilled water was added to give a 200 μL final volume. Then, 50 μL of 20% Na_2_CO_3_ (*w*/*v*) was added, and the sample was incubated for 1 h at 750 rpm and 25 °C. The absorbance of the reaction was then measured at 760 nm (Spectra max plus 384; Molecular Devices, CA, USA). The result is expressed as a percentage of the free phenol content, and the error bars were calculated as the average of three replicates.

## 3. Results

### 3.1. Cloning and Sequence Analysis of Irpex lacteus DyP

After sequencing of the gene coding for *I. lacteus* DyP from the fungal cDNA (three independent clones were analyzed to ensure proper sequence), the sequence was translated in the correct reading frame to obtain a 447 amino acids protein. This sequence did not include the signal peptide and presented an identity of 76% with the DyP from *Termitomyces albuminosus* (accession number AF468656) and 58% with the DyP from *B. adusta* (accession number CDN40127).

The alignment of the primary sequence of IrlacDyP with the sequences of DyPs from *A. auricula-judae* (GenBank accession number AFJ79723, 51% identity), *Pleurotus ostreatus* (DyP1 and Dyp4; JGI protein ID 1069077 and 62271, respectively, 41% identity), *T. versicolor* (XP_008039376.1, 44% identity), and *I. lacteus* D17 (Dyp1-4; AZJ17935.1, AZJ17936.1, AZJ17937.1, AVJ41190.1; 62%, 72%, 61%, and 82% identities, respectively) (Figure 2), revealed the presence of the defined conserved motif GXXDG, distal residues (Asp172, Arg335, Gly173, Leu360, and Phe362), heme proximal residues (His312, Val/Phe261, and Asp397), and radical-forming residues (Tyr340 and Trp380).

Figure 3 shows the phylogenetic analysis of the *I. lacteus* enzyme and other fungal DyPs based in their sequences, retrieved from Peroxibase (http://peroxibase.toulouse.inra.fr) and GenBank. Only an enzyme from *B. adusta* (10222 BaDyPrx01) belonged to DyP subfamily C, while all the other sequences were included in subfamilies D and B. The phylogram obtained with MEGA5 [35] includes four main clusters (I, II, IV, and V) and one small cluster (III). DyPs from basidiomycetes are included in clusters I–III, while *Ascomycota* enzymes are gathered into clusters IV and V (although cluster IV also includes two putative enzymes from *Postia placenta* and *Phlebiopsis gigantea*). IrlacDyP was included in cluster I, with type D enzymes from *B. adusta*, *A. auricula-judae*, and other *I. lacteus* strains and type C enzyme from *B. adusta*. All enzymes from type B included in the analysis were gathered into cluster V.

### 3.2. Production and Purification of the Recombinant Irpex lacteus DyP

The *I. lacteus* DyP coding sequence was cloned into pET28a and expressed in *E. coli* BL21 (DE3) pLysS as described in Materials and Methods. Small scale fermentations showed recombinant protein production as inclusion bodies, as reported for other fungal hemoperoxidases [36,37]. To obtain an active enzyme, it was necessary to optimize an in vitro refolding protocol. For that, a multifactorial analysis of the refolding mixture composition was performed in 96-well plates. The best refolding conditions (0.115 mg/mL protein, 1.6 mM GSSG, 0.16 M urea, 10 μM hemin, Britton-Robinson buffer pH 6.5, 4 °C) were scaled up to a volume of 3 L. This protocol was similar to that optimized for *A. auricula-judae* [38]. The maximum values for active protein were retrieved after 7 days of incubation (570 U/mL activity toward ABTS; 88 mU/mL toward RB19).

The refolded protein was concentrated, dialyzed, and applied onto a Resource Q anionic exchange column for purification (Figure 4). The protein was purified in only one chromatographic step, with a yield around 44%, which is much higher than that obtained for the native *I. lacteus* protein [21].

### 3.3. IrlacDyP Spectroscopic Characterization

IrlacDyP molecular mass was estimated to be around 50 kDa by SDS–PAGE (Figure 4B). This is in agreement with previous findings describing the native enzyme *M*_w_ after deglycosylating (51.11 kDa, [21]), and it is also similar to those reported for other fungal DyPs, such as those from *A. auricula-judae* [38], *P. ostreatus* [39], and other *I. lacteus* strains [40]. The isoelectric point of IrlacDyP characterized in this work was 4.09 (not shown), slightly higher than that found for the deglycosylated form of the native enzyme. Variations in the methodologies used (different ampholyte range) or the incomplete deglycosylation of the native enzyme could be the cause of this difference.

The Reinheitszahl value (A403nm/A280nm) calculated for the recombinant enzyme was 1.5, lower than that of the native protein but similar to those from other DyPs [41]. The pyridine hemochrome assay allowed calculation of the molar extinction coefficient at 403 nm (181,900 M^−1^cm^−1^), which is similar to that of the native enzyme [21]. At the resting state (RS), IrlacDyP showed a typical hemoperoxidase UV-visible spectrum (280–700 nm), indicating that protein folding and heme incorporation were correct (Figure 5A).

The Soret band and two charge transfer bands (CT1 and CT2) were observed at 403 nm, 633, and 504 nm, respectively. The spectral changes of the H_2_O_2_-activated enzyme were studied at pH 3.0 (optimum for RB19, RB5 and ABTS oxidation) and pH 5.5 (no activity with all substrates excepting Mn^2+^). At pH 3.0 (Figure 5B), the Soret band decreased and moved from 403 nm (black line) to 409 nm (dark green line). The CT2 band split up into two peaks at 528 and 556 nm, while CT1 remained unchanged. Then the spectrum returned to the RS without the appearance of additional or new maxima. These spectral changes are compatible with the formation of a reactive compound I (CI), as described for fungal (*A. auricula-judae* and *P. ostreatus*) and bacterial (*Thermomonospora curvata*) DyPs [12,39,42], including the quick self-reduction of compound II (CII) since its spectrum could not be observed. Similar IrlacDyP spectral changes were also observed when the resting enzyme reacted with H_2_O_2_ at pH 5.5 (Figure 5C), although at this pH the Soret band decreased and shifted its maximum from 403 to 416 nm (dark blue line). This Soret band displacement has been related to the Compound II-like formation in other fungal and bacterial peroxidases at pH 7–8 [12,39,42].

The circular dicroism spectrum of the recombinant DyP was recorded using 0.110 mg/mL protein in sodium acetate 20 mM (pH 5.5) (Figure 5D). An intense signal was detected, with 208 nm minimum lower than 222 nm valley, indicating predominance of α-helix and a maximum at 195 nm, corresponding to β-sheet portions [43]. Thus, an α-helix + β-sheet structure was detected, according to that found for other DyPs [12].

### 3.4. pH, Temperature, and H_2_O_2_ Effect on IrlacDyP Activity

The pH effect on IrlacDyP activity was studied in a range from 1.5–9.0 with ABTS, RB19, RB5, DMP, VA, and MnSO_4_ as substrates. The enzyme was more active at pH values between 1.5–6.0 (Figure 6A), but the optimum varied depending on substrates. DMP and VA oxidation were maximal at pH 2. The optimum pH for DMP oxidation was lower than that of the native enzyme and other fungal DyPs, while for VA it was in agreement with previous values [21,38,39,44]. RB19 and RB5 decolorization and ABTS oxidation were maximal at pH 3, although a second residual increment of RB5 decolorization was detected at pH 7. These values agree with those previously reported for other fungal DyPs [21,38,39,40]. RB19 decolorization was optimum at pH 3–4, slightly lower than previously reported values [38,39]. As in *P. ostreatus* DyP, the highest optimum pH (4.5) was observed with Mn^2+^ [39].

With respect to pH stability, IrlacDyP was incubated at pH 2–9 at 4 ºC and 25 °C, taking samples at different incubation times to measure residual activity against ABTS (Figure 6B,C). The enzyme was very stable between pH 3–6 at 4 °C, keeping 90% activity after 168 h incubation. The small decrease at pH 4 (~80% residual activity) may be explained because the pI of the enzyme was 4.1 and then the protein’s solubility could decrease at this pH. Enzyme stability was lower at 25 °C, retaining 90% activity between pH 5 and 6 but decreasing to 60% and 20% at pH 3 and 4, respectively, after 70 h incubation. Above and below those values, stability was drastically reduced after just a few hours of incubation. The native enzyme was more stable, probably because it was glycosylated [21].

Regarding thermal stability, the T_50_ index was adjusted to 48.3 ± 0.09 °C, after 10 min incubation between 30 and 65 °C (Figure 6D). This is much lower than the value obtained for the native enzyme and also lower than those reported for some recombinant MnPs from *Irpex* expressed in *E. coli* [15,21], but it is in the same range as that of DyP1 from *P. ostreatus* [39].

Inactivation by H_2_O_2_ in the absence of reducing substrate is a common feature to all heme peroxidases, either due to cofactor destruction, release of heme iron, or oxidation of amino acid side-chains [45]. To check H_2_O_2_ enzyme stability, IrlacDyP was incubated in the presence of different concentrations of this substrate, determining residual activity versus ABTS at different times (Figure 7). The protein became progressively inactivated with increasing incubation time and peroxide concentration, detecting the highest inactivation above 100,000 molar equivalents. Around 80% inactivation was observed with 3 × 10^3^ molar equivalents in 5 min, and the protein was completely deactivated after 10 min. The activity of the recombinant protein fell below 20% in 60 min of incubation at all the hydrogen peroxide concentrations tested, showing lower stability to this agent than the native enzyme. However, IrlacDyP was more stable than other DyP enzymes, such as that produced by *A. auricula-judae*, whose half-life in the presence of 3000 peroxide equivalents was 3.8 min [38].

The different stabilities determined for the native and recombinant DyP forms could be due to the lack of glycosylation of the enzyme produced in *E. coli* [36]. Thus, eukaryotic expression systems would be desirable for improving IrlacDyP properties. However, the production of recombinant active fungal heme peroxidases in eukaryotic system shows two main drawbacks: (i) the low levels of heterologous expression usually obtained and (ii) the hyperglycosylated protein usually obtained when yeasts are used as expression systems affecting the catalytic properties and stability of the recombinant peroxidases. Prokaryotic systems usually present well-known genetic backgrounds and fast growth rates, and complex protein can be accumulated as inclusion bodies [46], whose formation can be diminished by chaperones’ coexpression or continuous hemin addition, which has been successfully used for hemoperoxidase expression [47] to produce a more stable protein. Eukaryotic recombinant protein expression, on the other hand, eases protein folding and presents high secretion rates, though genetic manipulation can be more challenging, and sometimes hyperglycosylated proteins show lower recovery yields and specific activities [48,49]. In both cases, approaches such as protein engineering techniques could also serve to design a protein with improved stability, as has been reported for *P. eryngii* VP [50,51,52].

### 3.5. Kinetic Characterization of the Irpex lacteus Recombinant DyP

The kinetic parameters for the oxidation of different substrates were determined and compared with those from other fungal DyPs (Table 2). Native and recombinant DyP from *I. lacteus* showed similar kinetic constants toward RB19 and RB5, with higher affinity for H_2_O_2_ in the recombinant form. Native DyP seems to be more active on DMP and VA than the recombinant form. It is necessary to emphasize that although the oxidation of MnSO_4_ had not been described in the native IrlacDyP [21], oxidation of this compound by this enzyme has been detected in this study by using higher Mn^2+^ concentration (200 times more), although the native enzyme was less efficient oxidizing Mn^2+^ than the recombinant protein, due to its lower affinity.

Regarding other DyPs, RB19, and RB5 oxidation, the catalytic activity of IrlacDyP was similar to that shown by *A. auricula-judae* and *P. ostreatus* (DyP4) enzymes, although with higher affinity in the case of RB19. Nevertheless, the oxidation of ABTS, a low redox potential substrate, was more efficient with IrlacDyP than with these two enzymes. While *P. ostreatus* DyPs were not able to oxidize VA, the recombinant DyPs from *I. lacteus* and *A. auricula*-*judae* showed similar catalytic efficiencies toward that substrate. On the other hand, the catalytic efficiency of IrlacDyP versus MnSO_4_ was between those found for DyP1 and DyP4 from *P. ostreatus*, and much lower than that of *I. lacteus* F17, while the enzyme from *A. auricula-judae* could not oxidize this substrate.

To date, the recombinant DyPs from *Irpex* are the only fungal DyPs described to oxidize high redox potential dyes, such as VA and MnSO_4_, although research with DyPs from other sources is scarce. These compounds can be oxidized by LiPs and VPs, which have a Mn^+2^ oxidation site, a heme channel for low redox potential compounds, and a solvent-exposed catalytic Trp for high redox potential substrates [3]. Catalytic efficiencies of DyPs toward VA and MnSO_4_ are much lower than those from LiP and VP. In fact, the catalytic efficiency for VA and Mn^2+^ oxidation is 4- and 100-fold higher, respectively, in *P. eryngii* VP than in recombinant IrlacDyP, while for VA it is 400-fold higher in *P. chrysosporium* LiP [16].

### 3.6. Structural Model of the Irpex lacteus Recombinant DyP

In order to explain the IrlacDyP catalytic properties, a structural analysis was performed. The 3D molecular modeling of IrlacDyP was carried out using the crystal structure of the DyP from *B. adusta* (PDB 3MM3), chosen as the best template. The solvent access surface of the model obtained for this enzyme (Qmean value of −4.03) is shown in Figure 8A, where we can see the channel providing access to the heme group. The overall structure of the *I. lacteus* DyP is composed of two domains with ferredoxin-like fold that corresponds with that typical of DyPs (and rest of the members of the chlorite dismutase superfamily), with the heme cofactor in a central cavity located at the interface of the two domains. This fold is different from that of the classical ligninolytic peroxidases belonging to the peroxidase-catalase superfamily [53].

Figure 8B shows a detail of the IrlacDyP heme environment, which resembles that found in other DyPs. An aspartic (Asp172) and an arginine (Arg335) located over the heme plane (distal side) are expected to participate in the heterolytic cleavage of H_2_O_2_ and enzyme activation forming the compound I, as distal Asp168 and Arg332 have been described to do in AauDyP, or as distal arginine and histidine do in other classical fungal peroxidases [54]. At the other side of the heme plane (proximal side), His312 acts as fifth coordination position of the heme iron, being conserved in all basidiomycete DyPs and classical fungal peroxidases. Near the proximal histidine and aspartic (Asp397) residues, a phenylalanine occupies the position of Val253 in AauDyP or tryptophan in cytochrome C peroxidase [54]. In H_2_O_2_-activated cytochrome C peroxidase, a catalytic protein radical is located in this tryptophan residue [55,56], but no radicals have been described at this position in other peroxidases. In DyP enzymes, this place is occupied by both aromatic (phenylalanine) or aliphatic (valine, isoleucine or methionine) residues [6].

The oxidation of bulky and high redox potential substrates has been described in ligninolytic peroxidases (VPs and LiPs) and DyPs [12,32]. After H_2_O_2_ activation of the enzyme, solvent-exposed aromatic residues (tryptophans or tyrosines) are oxidized via long range electron transfer (LRET) pathways, generating protein radicals that are responsible for the oxidation of these substrates. Among the six tryptophan and five tyrosine residues of the mature IrlacDyP sequence, Trp380 occupies the same position of the catalytic Trp377 in AauDyP, and Tyr340 occupies the same position of Tyr337, both residues identified as protein radicals in the H_2_O_2_-activated AauDyP [12].

In contrast to the functional homologies mentioned above among ligninolytic peroxidases and DyPs, the only Mn^2+^ oxidation site currently characterized in the fungal DyP from *P. ostreatus* (PleosDyP4) differs from that of MnPs and VPs. The Mn^2+^ oxidation site of PleosDyP4 includes four acidic residues located at the surface of the enzyme, three aspartates and one glutamate (Asp-215, Asp-352, Asp-354, and Glu-345), implicated in Mn^2+^ binding, and one tyrosine (Tyr-339) that collects and transfers the subtracted electrons to the heme in a second LRET pathway. Neither of these residues are conserved in IrlacDyP, suggesting that a different Mn^2+^ oxidation site is present in this enzyme. Mn^+2^ oxidation sites described in bacterial C-type DyPs were not found to be conserved in IrlacDyP structure either [57,58]. On the other hand, the ability to oxidize Mn^2+^ in a resurrected basidiomycete D-type DyP has recently been reported, although a Mn^2+^-binding site was not found in this enzyme [59]. Thus, Mn^2+^ oxidation sites do not seem to be conserved in DyPs and MnP, neither between type C and type D DyPs, nor even between type D DyPs.

Regarding glycosylation, the native *I. lacteus* DyP displays 15 glycosylation sites predicted by in silico analysis of the mature protein (not shown). The lack of glycosylation at these positions in the recombinant IrlacDyP, expressed in *E. coli*, could be responsible for its different stability when compared with the native enzyme.

### 3.7. Transformation of Lignosulfonates

As explained above, one of the putative biological roles suggested for fungal DyPs is the degradation of lignin in plant biomass. However, the native polymer is a very complex substrate, and its direct oxidation has hardly been reported since it requires the synergistic effect of a complex ligninolytic machinery. One of the ways to study lignin degradation by oxidoreductases involves the use of models that emulate the structure of lignin but are less complex. Lignosulfonates (LS) are one of such models, as they are water-soluble compounds of low molecular weight derived from the paper industry [23]. The depolymerization of LS through oxidation by peroxidases and laccases is already known, but to date no data are available on their oxidation by DyP, although these catalysts are able to oxidize kraft lignin, phenolic lignin model dimers, and, in the case of fungal DyPs, even non-phenolic lignin model-dimers [13,15,18].

As a first approach to ascertain the possible catalytic role of *I. lacteus* DyP in lignin transformation, as well as its biotechnological interest, its ability to oxidize lignosulfonates was analyzed. In addition, the oxidation of lignosulfonates is an interesting biotechnological application itself since the oxidized units can be crosslinked, giving larger molecules that can be used as dispersants. The oxidation of lignosulfonates (5 g/L) was studied using different concentrations of enzyme (0.25 and 0.50 μM) in the presence of 14,000 H_2_O_2_ molar equivalents (3.5 and 7 mM), which were added in pulses during 24 h to avoid enzyme inactivation. The phenolic content of softwood LS did not substantially decrease with the enzymatic treatment (Figure 9A), but in hardwood LS a significant reduction of phenolic content was observed (Figure 9B). No clear differences between enzyme doses were detected after 2 h, although the decrease of LS phenolic content was evident after 24 h of treatment.

Differences found between softwood and hardwood LS could be due to various factors. While softwood LS composition is based on G-phenylpropanoid units, hardwood LS contains G- and S-phenylpropanoid units [2]. Thus, the *I. lacteus* DyP seems to be more efficient transforming S- than G-units. Another determining factor is the molecular size of the polymer, since hardwoods LS (5,500 Da) are smaller than softwoods LS (32,000 Da) and hence more accessible to enzyme degradation [32].

IrlacDyP ability for LS oxidation paves the way for its possible use in organic synthesis applications [60]. Additionally, it has been proved that native IrlacDyP succesfully improved cellulose digestibility of wheat straw when applied in combination with cellulolytic cocktails [21]. In addition to this application, IrlacDyP could be used for future industrial procceses already described for other DyPs, such as pulp bleaching, dye decolorization, and β-carotene degradation [7,61,62,63].

In this work, we described the production in *E. coli* of IrlacDyP, a recombinant form of the DyP of *I. lacteus*. This enzyme efficiently oxidizes typical substrates of DyPs such as azo and anthraquinonic dyes, and it is also capable of oxidizing Mn^2+^ and the non-phenolic VA, which has been described here for the first time in *I. lacteus* DyPs. In addition, we demonstrated that IrlacDyP modifies the phenolic content of LS, which points to a biological role of these enzymes related to the biodegradation of phenolic lignin and suggests their potential in applications aimed at the biotransformation of technical lignins.

## Figures and Tables

**Figure 1 jof-07-00325-f001:**
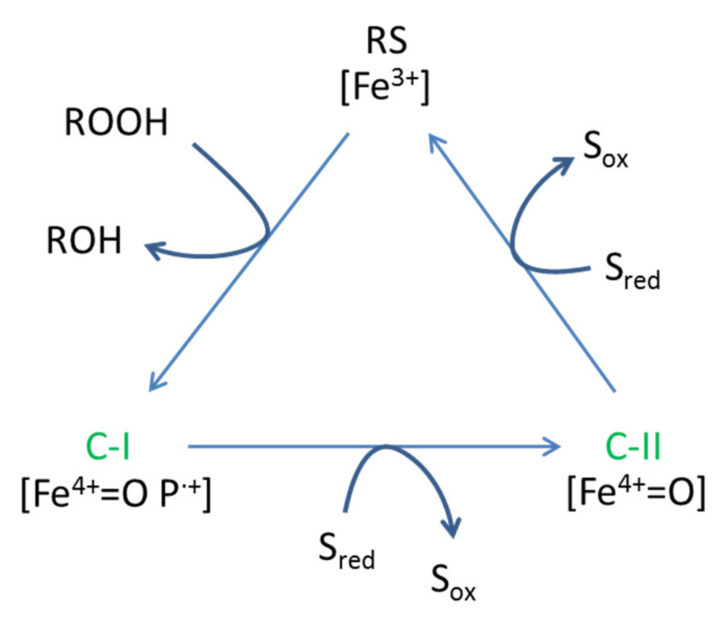
Catalytic cycle of peroxidases adapted from [9]. RS: resting state, CI: compound I; CII, compound II.

**Figure 2 jof-07-00325-f002:**
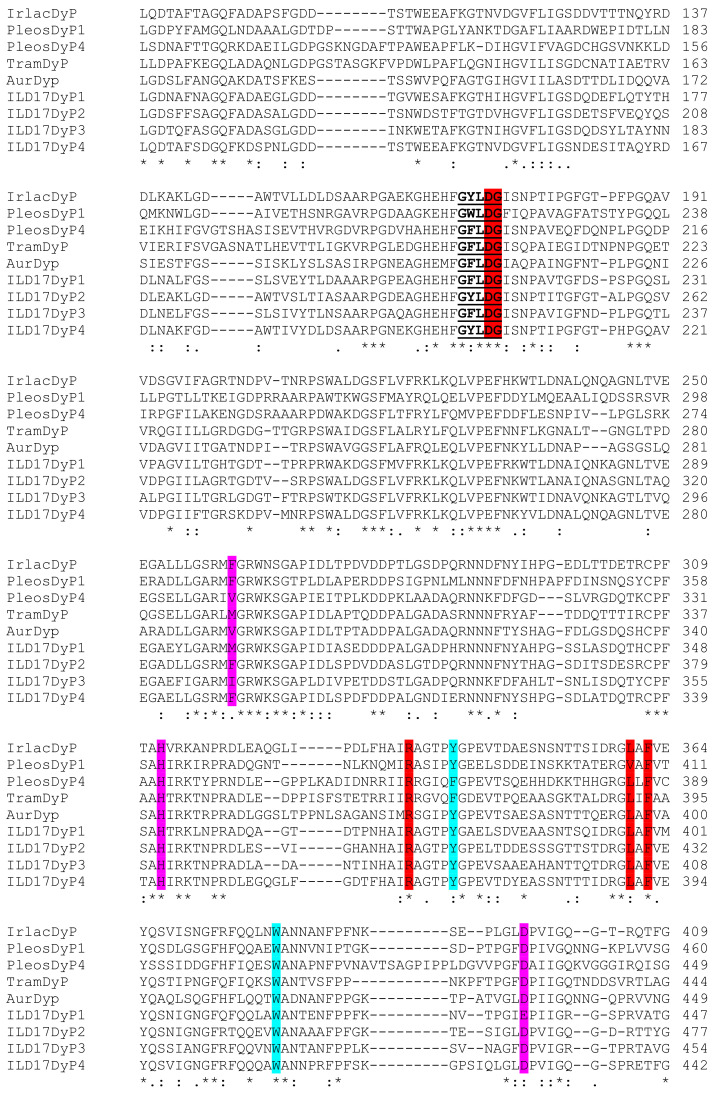
Sequence alignment including DyPs from *I. lacteus* (IrlacDyP), *A. auricula-judae* (AurDyp), *Pleurotus ostreatus* (PleosDyP1 and PleosDyP4), *T. versicolor* (TramDyP), and *I. lacteus* D17 (ILD17DyP1-4). GXXDG conserved motif is bold underlined; distal residues are highlighted in red; heme proximal residues are highlighted in purple, and putative radical-forming residues are highlighted in blue. Numbering corresponds to mature proteins. *- identical residues;:- conserved substitutions;. -semi-conserved substitutions.

**Figure 3 jof-07-00325-f003:**
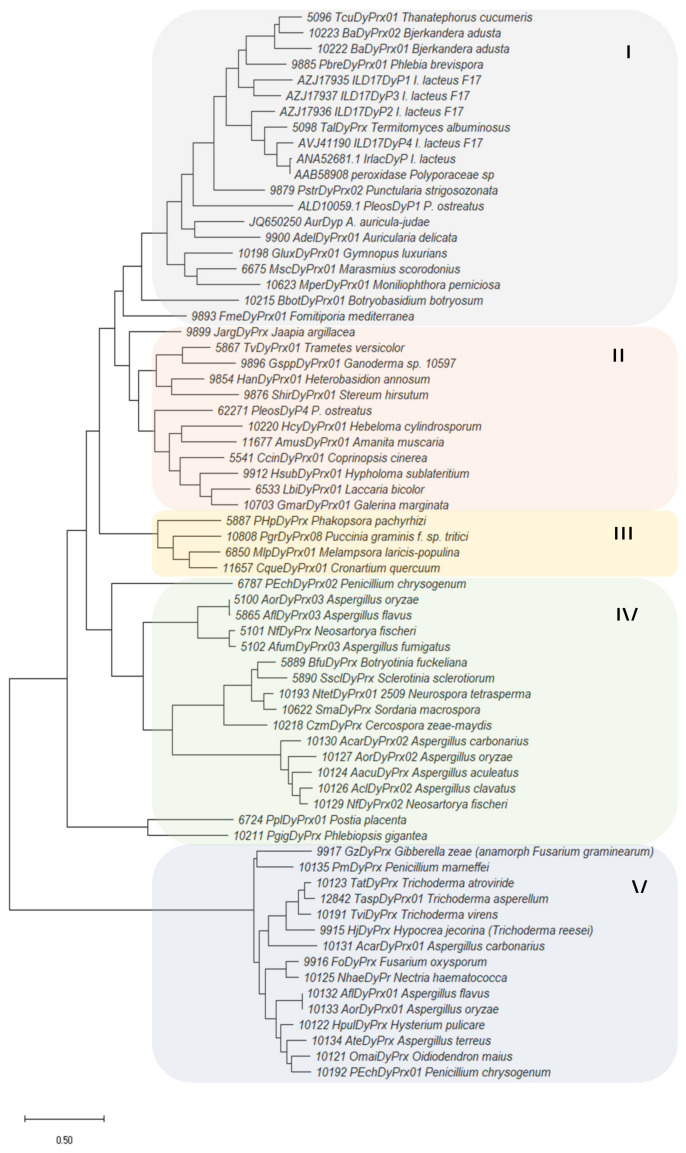
Phylogenetic analysis of the IrlacDyP and other fungal sequences from GenBank (sequences ALD10059.1, AVJ41190, AZJ17937, AZJ17936, AZJ17935, 62271, JQ650250, AAB58908, and ANA52681.1) and Peroxibase database (rest of sequences). Enzymes belonging to subfamilies B and D, and 10222 BaDyPrx01 from subfamily C, were included.

**Figure 4 jof-07-00325-f004:**
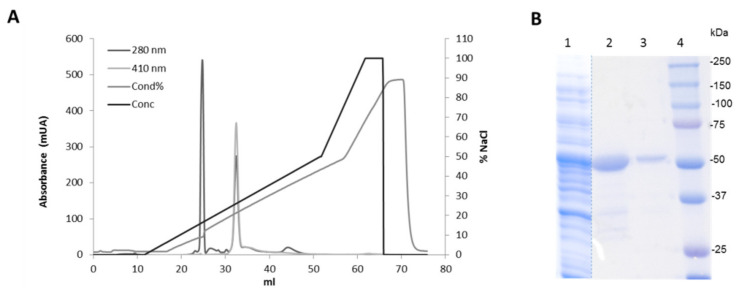
IrlacDyP purification. (**A**) Elution profile of the enzyme by anion exchange chromatography. (**B**) SDS–PAGE analysis of samples along the purification procedure. Lane 1, inclusion bodies (10 μg); lane 2, refolding mixture (5 μg); lane 3, purified IrlacDyP (1 μg); lane 4, molecular weight markers.

**Figure 5 jof-07-00325-f005:**
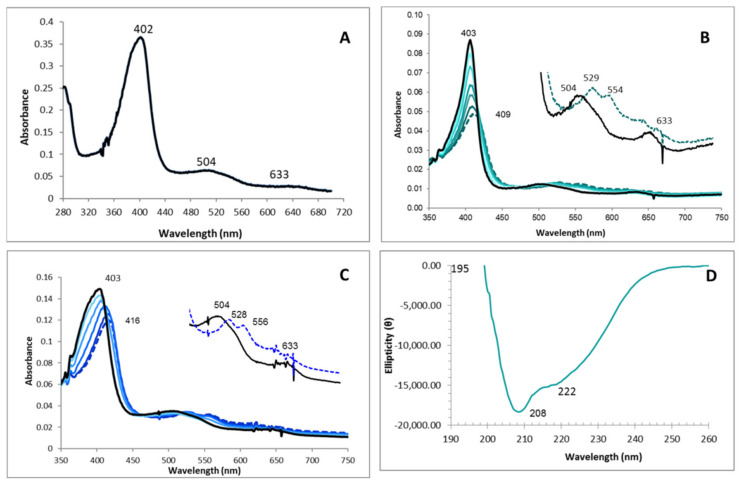
Spectroscopic characterization of *I. lacteus* DyP. (**A**) Electronic absorption spectrum (280–700 nm) of the purified IrlacDyP. (**B**) Electronic absorption spectra of 2.48 nM IrlacDyP before (black) and after addition of 5 H_2_O_2_ equivalents (~12.4 nM) at pH 3 (dark green line). (**C**) Electronic absorption spectra of 2.48 nM IrlacDyP before (black) and after addition of 5 H_2_O_2_ equivalents (~12.4 nM) at pH 5.5 (dark blue line). For (**B**,**C**), total spectrum was measured for 30 min at 5 s intervals. (**D**) Circular dichroism spectrum of the protein.

**Figure 6 jof-07-00325-f006:**
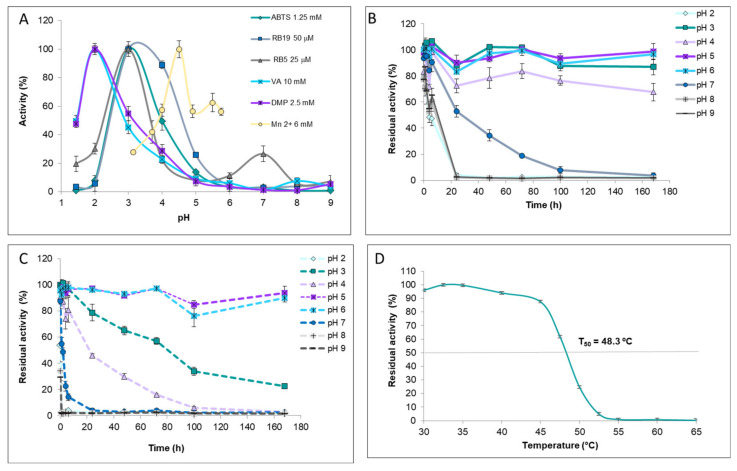
(**A**) Optimum pH for purified IrlacDyP. pH stability of IrlacDyP measured as residual activity with ABTS after incubation at different pH values at (**B**) 4 °C and (**C**) 25 °C. (**D**) T50 estimation for purified IrlacDyP.

**Figure 7 jof-07-00325-f007:**
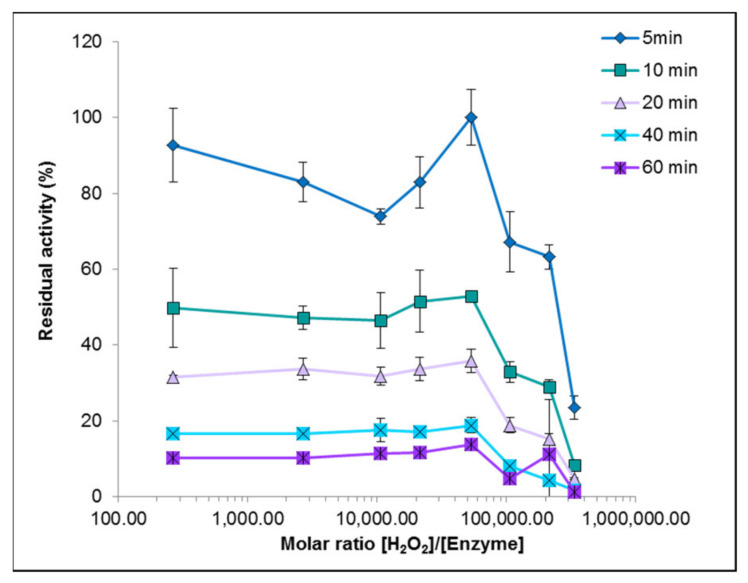
Hydrogen peroxide stability of IrlacDyP. Residual activity was determined with ABTS.

**Figure 8 jof-07-00325-f008:**
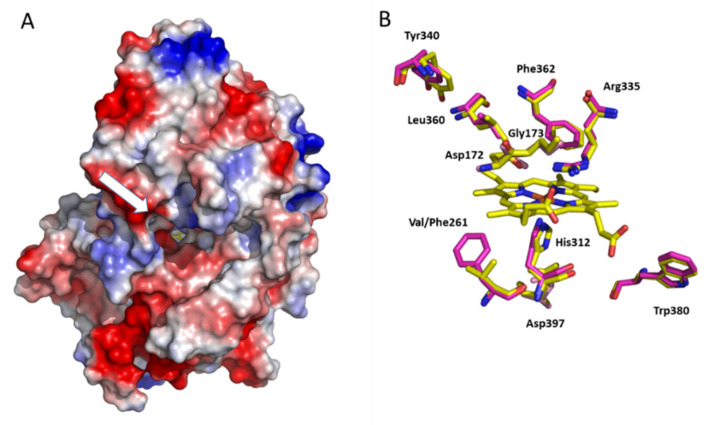
(**A**) Superimposition of the *I. lacteus* DyP (pink) and *A. auricula-judae* (PDB 4UZI) (yellow) structural models. (**B**) Heme cofactor environment residues, with Asp172 and Arg335 as putative residues involved in enzyme activation by hydrogen peroxide, His312 occupying the fifth coordination position of the heme iron; solvent-exposed Trp380 and Tyr340 (homologous to AauDyP catalytic Trp377 and protein radical forming Tyr337) are also shown. The white arrow indicates heme channel entrance.

**Figure 9 jof-07-00325-f009:**
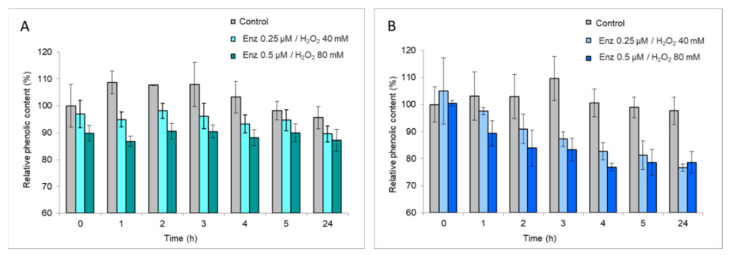
Analysis of phenolic content of softwood (**A**) and hardwood LS (**B**) in the presence of 0.25 μM and 0.50 μM of IrlacDyP and 40 and 80 mM H_2_O_2_, respectively, after 24 h of treatment. Controls without enzyme and H_2_O_2_ are included as grey bars.

**Table 1 jof-07-00325-t001:** Reaction conditions for kinetic characterization of IrlacDyP: wavelength (λ), extinction coefficients (Ɛ), substrate concentration ranges, and protein concentration.

Substrate	λ (nm)	Ɛ (cm^−1^ M^−1^)	(Substrate) (µM)	(Enzyme) (nM)	pH
H_2_O_2_ *	418	36,000	5–1960	1.25	3.0
ABTS	418	36,000	1–5000	1.25	3.0
DMP	469	27,500	2–2475	6.19	2.0
VA	310	9300	386–24,752	123.7	2.0
RB19	595	10,000	2.2–100	6.19	3.5
RB5	598	30,000	0.6–50	24.7	3.0
Mn^2+^	238	6500	23–23,760	61.9	4.5

* Kinetic constants for H_2_O_2_ enzyme activation were obtained using ABTS as reducing substrate.

**Table 2 jof-07-00325-t002:** Kinetic constants of recombinant IrlacDyP compared with native IrlacDyP [21] as well as other fungal DyPs from *A. auricula-judae* [38], *P. ostreatus* (PleosDyP1 and PleosDyP4) [39] and *I. lacteus* F17 (ILDyP4) [40] produced in *E. coli*. RB19: Reactive blue 19; RB5: Reactive black 5; VA: veratryl alcohol. Oxidation of Mn^2+^ by native IrlacDyP, included in the table (bold type), is reported for first time in this work.

		*Recombinant IrlacDyP*	*Native* *IrlacDyP*	*A. auricula-judae* DyP *	*Pleos*-DyP1	*Pleos*-DyP4	*IL-DyP4*
	*K*_m_ (μM)	1.5 ± 0.2	28.0 ± 2.6	123 ± 7	779 ± 69	787 ± 160	62 ± 11
*ABTS*	*k*_cat_ (s^−1^)	49.8 ± 1.3	224 ± 4.0	225 ± 3	208 ± 8	277 ± 24	8356 ± 747
	*k*_cat_/*K*_m_ (s^−1^ M^−1^)	(3.3 ± 0.4) × 10^7^	(8.0 ± 0.7) × 10^6^	(1.8 ± 0.09) × 10^6^	(2.7 ± 0.1) × 10^5^	(3.5 ± 0.5) × 10^5^	(1.3 ± 0.1) × 10^8^
	*K*_m_ (μM)	386. 0 ± 31.4	72.6 ± 9.5	703 ± 60	311000 ± 3800	126 ± 21	58 ± 3
*DMP*	*k*_cat_ (s^−1^)	94.5 ± 2.3	70.9 ± 2.1	120 ± 3	64 ± 3	268 ± 24	4896 ± 131
	*k*_cat_/*K*_m_ (s^−1^ M^−1^)	(2.4 ± 0.2) × 10^5^	(9.7 ± 0.1) × 10^5^	(1.7± 0.2) × 10^5^	(2.1 ± 0) × 10^3^	(2.1 ± 0.3) × 10^6^	(8.4 ± 0.2) × 10^7^
	*K*_m_ (μM)	6880. 9 ± 1136.9	3610.0 ± 211.0	-	-	-	(2.1 ± 0.98) × 10^4^
*VA*	*k*_cat_ (s^−1^)	3.9 ± 0.3	2.70 ± 0.1	-	-	-	108 ± 47
	*k*_cat_/*K*_m_ (s^−1^ M^−1^)	(5.7 ± 0.1) × 10^2^	(8.3 ± 0.0) × 10^2^	(1.0± 0) × 10^2^	-	-	(5.2 ± 2.3) × 10^3^
	*K*_m_ (μM)	9.0 ± 1. 3	13.5 ± 1.6	90 ± 10	45 ± 7	82 ± 13	133 ± 34
*RB19*	*k*_cat_ (s^−1^)	58.5 ± 2.1	79.9 ± 3.2	224 ± 10	5 ± 0.4	152 ± 13	5345 ± 921
	*k*_cat_/*K*_m_ (s^−1^ M^−1^)	(6.5 ± 1.0) × 10^6^	(5.9 ± 0.5) × 10^6^	(2.4 ± 0.2) × 10^6^	(1.1 ± 0.1) × 10^5^	(1.9 ± 0.1)10^6^	(4.0 ± 0.7) × 10^7^
	*K*_m_ (μM)	11.9 ± 2.1	11.2 ± 0.9	16 ± 2	-	5.7 ± 0.4	159 ± 61
*RB5*	*k*_cat_ (s^−1^)	12.9 ± 1.1	11.9 ± 0.4	4.8 ± 0.2	0	5.3 ± 0.8	267 ± 96
	*k*_cat_/*K*_m_ (s^−1^ M^−1^)	(1.1 ± 0.2) × 10^6^	(1.1 ± 0.05) × 10^6^	(3.1 ± 0.2) × 10^5^	0	(1.1 ± 1.0) × 10^6^	(1.7 ± 0.6) × 10^6^
	*K*_m_ (μM)	39.5 ± 7.7	79.5 ±11.7	-	-	-	163 ± 20.8
*H_2_O_2_*	*k*_cat_ (s^−1^)	145.3 ± 8.5	419.0 ± 18.8	-	-	-	(1.2 ± 0.08) × 10^4^
	*k*_cat_/*K*_m_ (s^−1^ M^−1^)	(3.68 ± 0.7) × 10^6^	(5.3 ± 0.6) × 10^6^	-	-	-	(7.6 ± 0.5) × 10^7^
	*K*_m_ (μM)	82.0 ± 24.9	**293.8 ± 39.9**	-	2780 ± 440	286 ± 33	2687 ± 463
*Mn^2+^*	*k*_cat_ (s^−1^)	1.2 ± 0.1	**1.8 ± 0.1**	-	10 ± 1	56 ± 2	806 ± 94
	*k*_cat_/*K*_m_ (s^−1^ M^−1^)	(1.41 ± 0.43) × 10^4^	**(6.1 ± 0.85) × 10^3^**	-	(4.0 ± 0) × 10^3^	(2.0 ± 0.2) × 10^5^	(3.0 ± 0.4) × 10^5^

* DMP and RB19 values of high-turnover site (at Trp377) [38]. - not determined. *K*_m_, affinity constant; *k*_cat_, turnover number_;_
*k*_cat_ /*K*_m_, catalytic efficiency; ABTS, 2,2′-Azino-bis(3-ethylbenzothiazoline-6-sulfonic acid) diammonium salt; DMP, 2,6-dimethoxyphenol; VA, veratryl alcohol; RB19,Reactive Blue 19; RB5, Reactive Black 5; H_2_O_2_, hydrogen peroxide; Mn^2+^, manganese ion.

## Data Availability

Not applicable.

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
