# Peer review of "Characterization of a Dye-Decolorizing Peroxidase from *Irpex lacteus* Expressed in *Escherichia coli*: An Enzyme with Wide Substrate Specificity Able to Transform Lignosulfonates"

_jof, 2021, doi:10.3390/jof7050325_

Round 1

Reviewer 1 Report

The paper entitled “Characterization of a dye-decolorizing peroxidase from Irpex lacteus expressed in Escherichia coli: an enzyme with wide substrate specificity able to transform lignosulfonates by de Eugenio et al., reports the cloning and expression in E. coli of I. lacteus DyP, and discusses properties of the recombinant protein (IrlacDyP) including the ability of the recombinant protein to oxidize lignosulfonates. The work is very solid with clear and interesting results. Some points to be considered are listed below.

1. I would strongly suggest the authors to rewrite introduction section with an emphasis on DyPs. In the current version of the paper, description starts with ligninolytic peroxidase LiP, MnP and VP including their EC numbers, while no EC number was provided for DyP, at the same time introducing four superfamilies of heme peroxidases although there are more in Peroxibase, so it’s a bit confusing for readers. It would be easier for the readers to follow if you can start with Dye-decolorizing peroxidases (DyPs) as a new family of heme peroxidases that have been classified according to sequence similarity into four types, A, B, C, and D, since D type is mentioned in the abstract and later in the results, also a bit inconsistently. It should be described in the Introduction, since the type D DyPs are exclusively found in fungi. It would be useful to describe how representatives from classes A, B, C, and D show similar tertiary structures (a ferredoxin-like fold), although they do not share high sequence similarity. Also include description of active site structure, with a proximal histidine as a strong ligand of the iron, and an arginine in the distal heme cavity, but the distal histidine, essential for the plant peroxidases, is absent in DyPs. Instead, there is a conserved aspartate, substituted by a glutamate in some rare cases, which is part of the DyP-typifying sequence fingerprint motif GXXDG.

2. Be very careful with statement that lacteus DyPs are the only enzymes able to oxidize high redox potential dyes, VA and Mn+2. The fact that some DyPs have not been tested for their activities against all of those substrates (for example DyP4 from Pleurotus eryngii Appl Microbiol Biotechnol (2015) 99:8927–8942), does not mean that those are not able to oxidize it. Furthermore it is known that even if some fungal DyPs are also able to oxidize veratryl alcohol and nonphenolic lignin model dimers, their activity is too low, so comparison of efficiency against VA should be compared with the literature data not only for DyPs, but with other ligninilytic peroxidases as well.

3. When describing the results of phylogenetic analysis of the IrlacDyP you explained that only an enzyme from adusta (10222 BaDyPrx01) belonged to DyP subfamily C, while all the other sequences were included in subfamilies D and B. Than at the end you concluded that IrlacDyP was included in cluster I, with type D enzymes from B. adusta, A. auricula-judae and other I. lacteus strains. Please correct that and explain the meaning of five clusters in relation to the DyP types.

4. Line 487: It is worth noting that the recombinant protein was able to oxidize Mn2SO4 although it was not previously described in native DyP. This needs to be further explained. Furthermore, lines 546-552: amongst other you claimed that in contrast to the functional homologies mentioned above among ligninolytic peroxidases and DyPs, the only Mn2+ oxidation site currently characterized in the fungal DyP from ostreatus (PleosDyP4) differs from that of MnPs and VPs. The Mn2+ oxidation site of PleosDyP4 includes four acidic residues located at the surface of the enzyme, three aspartates and one glutamate (Asp-215, Asp-352, Asp-354 and Glu-345), implicated in Mn2+binding, and one tyrosine (Tyr-339) that collects and transfers the subtracted electrons to the heme in a second LRET pathway. Neither of these residues are conserved in IrlacDyP. However, there are other, different DyPs with described small Mn2+-binding pocket near the edge of heme, such as R. jostii RHA1 DypB heme residue, Asn246, and DyP2, both solved in complex with Mn2+ (DyP2 (PDB: 4G2C) from Amycolatopsis sp 75iv2; and N246A variant of DypBRHA1 (D, PDB: 3HOV). Can you elaborate more about Mn2+ oxidizing ability including those as well?

5. Please correct the title for the Figure 8. A) It is the model obtained for this enzyme that is shown, not the alignment.

6. Lines 593-594: This enzyme efficiently oxidizes typical substrates of DyPs like azo and anthraquinonic dyes, and it is also capable of oxidizing Mn2+ and the nonphenolic VA, which is described for the first time in a DyP. Please rephrase this sentence, because this is not the first time that this is described in a DyP.

7. Please carefully check your reference, for example I couldn’t find where you have cited refrence 46 in the main text: Qin, X.; Sun, X.; Huang, H.; Bai, Y.; Wang, Y.; Luo, H.; Yao, B.; Zhang, X.; Su, X. Oxidation of a non-phenolic lignin model compound by two Irpex lacteus manganese peroxidases: Evidence for implication of carboxylate and radicals. Biofuels 2017.

Author Response

The paper entitled “Characterization of a dye-decolorizing peroxidase from Irpex lacteus expressed in Escherichia coli: an enzyme with wide substrate specificity able to transform lignosulfonates by de Eugenio et al., reports the cloning and expression in E. coli of I. lacteus DyP, and discusses properties of the recombinant protein (IrlacDyP) including the ability of the recombinant protein to oxidize lignosulfonates. The work is very solid with clear and interesting results. Some points to be considered are listed below.

  1. I would strongly suggest the authors to rewrite introduction section with an emphasis on DyPs. In the current version of the paper, description starts with ligninolytic peroxidase LiP, MnP and VP including their EC numbers, while no EC number was provided for DyP, at the same time introducing four superfamilies of heme peroxidases although there are more in Peroxibase, so it’s a bit confusing for readers. It would be easier for the readers to follow if you can start with Dye-decolorizing peroxidases (DyPs) as a new family of heme peroxidases that have been classified according to sequence similarity into four types, A, B, C, and D, since D type is mentioned in the abstract and later in the results, also a bit inconsistently. It should be described in the Introduction, since the type D DyPs are exclusively found in fungi. It would be useful to describe how representatives from classes A, B, C, and D show similar tertiary structures (a ferredoxin-like fold), although they do not share high sequence similarity. Also include description of active site structure, with a proximal histidine as a strong ligand of the iron, and an arginine in the distal heme cavity, but the distal histidine, essential for the plant peroxidases, is absent in DyPs. Instead, there is a conserved aspartate, substituted by a glutamate in some rare cases, which is part of the DyP-typifying sequence fingerprint motif GXXDG.

We agree with the reviewer and the introduction has been rearranged, as suggested.

  1. Be very careful with statement that lacteus DyPs are the only enzymes able to oxidize high redox potential dyes, VA and Mn+2. The fact that some DyPs have not been tested for their activities against all of those substrates (for example DyP4 from Pleurotus eryngii Appl Microbiol Biotechnol (2015) 99:8927–8942), does not mean that those are not able to oxidize it. Furthermore it is known that even if some fungal DyPs are also able to oxidize veratryl alcohol and nonphenolic lignin model dimers, their activity is too low, so comparison of efficiency against VA should be compared with the literature data not only for DyPs, but with other ligninilytic peroxidases as well.

We have modified the paragraph about the uniqueness of I. lacteus enzymes, since, as the reviewer pointed, the absence of data does not mean that other DyPs could not oxidize these substrates. Paragraph in lines 507-509 has been rewritten.

  1. When describing the results of phylogenetic analysis of the IrlacDyP you explained that only an enzyme from adusta (10222 BaDyPrx01) belonged to DyP subfamily C, while all the other sequences were included in subfamilies D and B. Than at the end you concluded that IrlacDyP was included in cluster I, with type D enzymes from B. adusta, A. auricula-judae and other I. lacteus strains. Please correct that and explain the meaning of five clusters in relation to the DyP types.

We apologize for the mistake. Indeed, 10222 BaDyPrx01 is the only class C DyP include in the tree, a sentenced has been included to clarify this point. All enzymes from type B were gathered in cluster V, as it has been included in the text (lines 367-369).

  1. Line 487: It is worth noting that the recombinant protein was able to oxidize Mn2SO4 although it was not previously described in native DyP. This needs to be further explained.

When native IrlacDyP was described and characterized [1], MnSO4 oxidation was not detected in the conditions assayed (maximal MnSO4 concentration was 200 times lower than in the present work). In this manuscript, we have decided to retest IrlacDyP activity on MnSO4 using improved conditions. In fact, native IrlacDyP is also able to oxidize Mn, but at lesser extent than recombinant protein. The new kinetic constants have been added to Table 2 and the text has been modified accordingly (lines 493-497).

Furthermore, lines 546-552: amongst other you claimed that in contrast to the functional homologies mentioned above among ligninolytic peroxidases and DyPs, the only Mn2+ oxidation site currently characterized in the fungal DyP from ostreatus (PleosDyP4) differs from that of MnPs and VPs. The Mn2+ oxidation site of PleosDyP4 includes four acidic residues located at the surface of the enzyme, three aspartates and one glutamate (Asp-215, Asp-352, Asp-354 and Glu-345), implicated in Mn2+binding, and one tyrosine (Tyr-339) that collects and transfers the subtracted electrons to the heme in a second LRET pathway. Neither of these residues are conserved in IrlacDyP. However, there are other, different DyPs with described small Mn2+-binding pocket near the edge of heme, such as R. jostii RHA1 DypB heme residue, Asn246, and DyP2, both solved in complex with Mn2+ (DyP2 (PDB: 4G2C) from Amycolatopsis sp 75iv2; and N246A variant of DypBRHA1 (D, PDB: 3HOV). Can you elaborate more about Mn2+ oxidizing ability including those as well?

We really appreciate this suggestion. In fact, IrlacDyP and R. jostii DyP 3D structural model superposition showed no clear similarities that could allow the search for homologous residues. When comparing IrlacDyP and Amycolatopsis sp. DyP (PDB 4G2C), we could detect that E273 from Amycotopsis sp. DyP is occupying a close position to IrlacDyP E251. However, the proposed Mn pocket in 4G2C is not found in IrlacDyP structure. These data have not been included in the manuscript.

The sentence: “Mn+2 oxidation sites described in bacterial DyPs were not found to be conserved in IrlacDyP structure either” has been included in the manuscript, lines 570-575.

  1. Please correct the title for the Figure 8. A) It is the model obtained for this enzyme that is shown, not the alignment.

Thank you for the correction. The title has been rearranged to “Superimposition of the I. lacteus DyP (pink) and A. auricula-judae (PDB 4UZI) (yellow) structural models”

  1. Lines 593-594: This enzyme efficiently oxidizes typical substrates of DyPs like azo and anthraquinonic dyes, and it is also capable of oxidizing Mn2+ and the nonphenolic VA, which is described for the first time in a DyP. Please rephrase this sentence, because this is not the first time that this is described in a DyP.

Sentence has been rephrased to “which has been described for the first time in I. lacteus DyPs” (Line 623)

  1. Please carefully check your reference, for example I couldn’t find where you have cited refrence 46 in the main text: Qin, X.; Sun, X.; Huang, H.; Bai, Y.; Wang, Y.; Luo, H.; Yao, B.; Zhang, X.; Su, X. Oxidation of a non-phenolic lignin model compound by two Irpex lacteus manganese peroxidases: Evidence for implication of carboxylate and radicals. Biofuels 2017.

References have been carefully checked.

Reviewer 2 Report

In this manuscript, the authors have biochemically and technologically characterized the dye-decolorizing peroxidase from Irpex lacteus, purified after the expression of the corresponding gene in Escherichia coli. Moreover, a possible biological role of this protein in the degradation of phenolic lignin for transformation of softwood and hardwood lignosulfonates is also indicated

The results of the biochemical study, which has been performed in an excellent way, clearly described the protein of the above enzyme. Moreover, the phylogenetic analysis revealed its homology with other type of D-Dy peroxidase from basidiomycetes.  This work was well done and the methods were appropriate used. I think that it is a worthy study with interesting information to report.

However, some improvements could be made to the manuscript before publication:

-The introduction is too verbose and it should be reduced

-Line 302-303. How many independent sequences have been carried out in order to obtain the gene coding?

-The discussion is not reported as section: However, it could be boosted by discussing the following articles:

Zitare et al., (2020). Mutational and structural analysis of an ancestral fungal dye decolorizing peroxidase. The FEBS Journal.

Pech-Canul et al., (2020). Functional Expression and One-Step Protein Purification of Manganese Peroxidase 1 (rMnP1) from Phanerochaete chrysosporium Using the E. coli-Expression System. International journal of molecular sciences, 21(2), 416.

Behrens et al., (2016). Comparative cold shock expression and characterization of fungal dye-decolorizing peroxidases. Applied biochemistry and biotechnology, 179(8), 1404-1417.

Lin et al., (2018). High yield production of fungal manganese peroxidases by E. coli through soluble expression, and examination of the activities. Protein expression and purification, 145, 45-52.

-Moreover, the Authors should: 1) describe the plus and minus of a prokaryotic expression system versus the eukaryotic ones for fungal peroxidase production; 2) highlight the potential industrial application of the heterologous I. lacteus peroxidase.

Author Response

Reviewer 2:

In this manuscript, the authors have biochemically and technologically characterized the dye-decolorizing peroxidase from Irpex lacteus, purified after the expression of the corresponding gene in Escherichia coli. Moreover, a possible biological role of this protein in the degradation of phenolic lignin for transformation of softwood and hardwood lignosulfonates is also indicated

The results of the biochemical study, which has been performed in an excellent way, clearly described the protein of the above enzyme. Moreover, the phylogenetic analysis revealed its homology with other type of D-Dy peroxidase from basidiomycetes.  This work was well done and the methods were appropriate used. I think that it is a worthy study with interesting information to report.

However, some improvements could be made to the manuscript before publication:

-The introduction is too verbose and it should be reduced

Introduction has been reduced as suggested.

-Line 302-303. How many independent sequences have been carried out in order to obtain the gene coding?

Three different clones were sequenced to ensure that gene sequencing was accurate.  A sentence was included in the manuscript: “three independent clones were analyzed to ensure proper sequence” (line 297-298).

-The discussion is not reported as section: However, it could be boosted by discussing the following articles:

Zitare et al., (2020). Mutational and structural analysis of an ancestral fungal dye decolorizing peroxidase. The FEBS Journal.

Pech-Canul et al., (2020). Functional Expression and One-Step Protein Purification of Manganese Peroxidase 1 (rMnP1) from Phanerochaete chrysosporium Using the E. coli-Expression System. International journal of molecular sciences, 21(2), 416.

Behrens et al., (2016). Comparative cold shock expression and characterization of fungal dye-decolorizing peroxidases. Applied biochemistry and biotechnology, 179(8), 1404-1417.

Lin et al., (2018). High yield production of fungal manganese peroxidases by E. coli through soluble expression, and examination of the activities. Protein expression and purification, 145, 45-52.

Articles have been included in the discussion. We are grateful for reviewer recommendation.

-Moreover, the Authors should: 1) describe the plus and minus of a prokaryotic expression system versus the eukaryotic ones for fungal peroxidase production;

Eukaryotic expression systems would be desirable for the production of recombinant active fungal heme peroxidases. In fact, there are different examples in the literature in which filamentous fungi and yeasts have been used for this purpose, including the use of constitutive and inducible promoters, protease deficient strains, overexpression of chaperons, and the introduction of an external source of heme. However, there are two main drawbacks related to the use of these hosts for the study/production of fungal peroxidases. Low levels of heterologous expression are usually obtained, and the enzymes are hyperglycosylated when yeasts are used as expression systems affecting the catalytic properties and stability of the recombinant peroxidases.

In fact, bacteria are a good alternative to the above eukaryotic systems at lab scale, even though in most cases the recombinant fungal hemeperoxidases accumulate in inclusion bodies and it is necessary to optimize an in vitro refolding protocol to obtain active enzymes. In these cases the prokaryotic systems provide advantages compared to the eukaryotic heterologous expression, including i) higher amounts of recombinant proteins obtained in shorter times due to the rapid growth of bacteria; ii) the recombinant protein purification is done in only one chromatographic step (when they are produced in inclusion bodies and are in vitro reactivated); and iii) these enzymes are easy to crystallize due to the absence of glycosilation (micro-heterogeneity of glycosilated proteins usually make difficult their crystallization) and they are an optimal material for structure-function studies.

A sentence has been included in manuscript (lines 476-487).

2) highlight the potential industrial application of the heterologous I. lacteus peroxidase.

A sentence has been included (lines 613-618).

Reviewer 3 Report

The manuscript submitted by de Eugenio et al. reports on the cloning and molecular characterization of a DyP-type peroxidase derived from the white-rot basidiomycete Irpex lacteus. State-of-the-art methods have been employed and the data are relevant and interesting. However, as peroxidases with highly similar characteristics have been described in detail e.g. from other Irpex species (e.g. ref. 41), A. auricula-judae and P. ostreatus, the overall novelty of the manuscript is somewhat limited.

While the science is sound and generally deserves publication, the overall technical quality of the writing needs to be improved. Most of the chemical formulas have to be corrected (e.g. Mn2SO4, H2O2, Mn2+, Mn+2. etc. ), and Latin expressions (in vitro, etc). and species names (A. auricula judae, E. coli, etc.) need to be italicized.

Additional minor topics:

Figure 4: how much protein was loaded on the gel?

Figure 8: the legend does not fit to the figure

The symbol for “micro” is corrupted

Author Response

Not applicable